# Analysis of Mechanical and Thermal Properties of Polymer Materials Derived from Recycled Overprinted Metallized PP Films

**DOI:** 10.3390/ma17081739

**Published:** 2024-04-10

**Authors:** Tomasz Stachowiak, Przemysław Postawa, Marcin Chmielarz, Dominik Grzesiczak

**Affiliations:** 1Department of Technology and Automation, Faculty of Mechanical Engineering and Computer Science, Częstochowa University of Technology, Al. Armii Krajowej 21, 42-201 Częstochowa, Poland; przemyslaw.postawa@pcz.pl; 2Granulat-Bis Company, ulica Hallera 8A, 42-202 Częstochowa, Poland; m.chmielarz@granulatbis.com.pl (M.C.); dominikg88@gmail.com (D.G.)

**Keywords:** recycling, polymeric materials, layered films, metallized films

## Abstract

Polymer materials and their composites are one of the most frequently used materials in the packaging and food industries. This applies to both disposable and reusable packaging, layered films with barrier properties, as well as densely overprinted polymer films and metallized food wrap films. According to statistical data from Plastics Europe, approximately 40% of processed thermoplastics are used to produce packaging, including single- and multi-layer film packaging. Growing requirements and new EU directives require the use of recycled materials in new products, which is not easy because the properties of recyclates may differ significantly from those of the primary materials with which the former are mixed. This work attempts to analyze the properties of the primary material used to produce a film using the casting method in comparison with the industrial recyclate obtained by the processing of film made of the primary material and then overprinted and metallized. The process of obtaining re-granulates and preparing test samples was presented, and the mechanical, structural, and thermal properties of the tested materials were compared. The conducted research and the obtained results demonstrated the advisability of conducting advanced mechanical recycling, which leads to obtaining re-granulates with repeatable processing properties and thermal and mechanical properties comparable to the original material despite the impurities they contain.

## 1. Introduction

Films used in the food industry mainly belong to the group of thermoplastic polymer materials; these are polyethylene or polypropylene films [1]. In many cases, they are multi-layered or combined with other types of materials in order to gain properties that are impossible to obtain when using single-layer or single-material foils (including metallized and overprinted films) [2,3,4]. The introduction of new materials and multi-layer solutions is to ensure the unique properties of the film, including vapor permeability or the blocking of the transmission of certain gases or migration of biological substances (e.g., fat or bacteria, but also to improve mechanical, structural, or optical properties). In addition, modern solutions—such as overprint, labels, or decorations, introducing attractive marketing features that support sales—have an adverse impact on the possibilities of plastic packaging management when it turns into waste [5,6,7,8,9]. This is mainly related to the way the packaging is recycled and reused. Introducing multi-layer wrappings or films combined with other materials (including metals) makes it difficult or, in many cases, impossible to properly recycle them. This type of waste has so far been excluded from this stream and was used as alternative refuse-derived fuels (RDFs) after a process of preparing mixed waste with high calorific value (above 18 MJ/kg). For higher calorific value, the mixture may contain industrial waste. Therefore, only energy recovery was carried out for this type of waste, without the possibility of recycling in accordance with the principles of the circular economy [8,9,10,11].

Processing and recycling technologies have been intensively developed in recent years to reuse polymer waste characterized by a high degree of contamination [12,13,14,15,16,17]. This type of waste also includes food wrap films. The question may arise as to why to invest such significant amounts of money in materials and technologies intended for the reprocessing of contaminated thermoplastic materials. It should be taken into account that, currently, nearly 40% of the world’s production of polymer materials is consumed by the production of packaging, including films, whose product life cycle ranges from several days to several weeks. The lack of appropriate tools for the reuse of this type of waste makes it difficult to reduce CO_2_ emissions and results in the irreversible loss of non-renewable resources, such as the hydrocarbons from which thermoplastics are made. The European market today consumes over 58 Mt of thermoplastics per year, as shown in Figure 1 [18].

A thorough analysis of the issue of the reuse of waste films and pliable/flexible packaging was carried out by Horodytska et al. (2018). The authors systematized the division of waste received in the post-consumer waste stream and suggested some methods and possibilities for their disposal. The authors concluded that it is necessary to correlate the design of the recycling process with the broadly understood issue of the life cycle assessment as well as the collection and sorting of waste [19]. Similar problems were addressed by Cabrera et al. (2022). The authors analyzed the market for the reprocessing of multi-layer post-consumer waste films, taking into account not only mechanical recycling but also chemical recycling and energy recovery. The authors indicated many paths and possible ways of recycling plastic waste. Moreover, they concluded that the processing of multi-layer films, with each layer made of a different material, becomes extremely demanding. As they demonstrated, the chemical recycling of thermoplastic polymers seems to be extremely promising [20]. Unlike Cabrera et al., Sanetuntikul et al. (2023) analyzed the impact of waste from metallized films used as an additive (filler) to polypropylene processed by injection molding technology. The authors indicated the construction, furniture, and agricultural industries as possible areas of application of composites, provided, however, that the material does not come into contact with food [21,22]. Similar research and applications of recycled waste were also described by Bhogayata et al. (2019). The authors focused on the possibility of using properly prepared waste metallized films as fibers added to concrete in order to improve its properties. As their tests showed, the mechanical properties and impact strength were significantly improved thanks to the use of metallized film fibers. Moreover, the use of this type of fiber significantly prevented the formation of wide cracks. As the authors proved, the disposal of this type of waste by mixing it with concrete is very desirable from an environmental point of view and should be taken into account in life cycle assessments [10,22]. Many researchers also draw attention to a different approach to the issue of recycling. The first mentions of attempts to chemically recycle waste in the context of managing packaging waste and film appeared as early as 1979 in the study by Murphy et al. (1979). Their tests showed a progressive degradation process in recycled materials, but the impact of the degradation was not strong enough to significantly affect the material properties. A continuation and extension of this work is the publication by Walker et al. (2020). The authors presented a broad analysis of the possibilities of using solvent-targeted recovery and precipitation (STRAP) technology for the recycling and recovery of polymer materials, including those from multi-layer films. The authors’ method included the process of selecting solvents for recycled materials and a method for recovering individual fractions. This is an extremely interesting approach that enables the elimination of imperfections in the mechanical recycling process. At the moment, however, mechanical recycling, with its varieties and modifications, is the most popular and efficient process enabling the management of thermoplastic waste [23,24,25,26,27,28].

The aim of this research was to use the mechanical recycling process to reuse waste from the industrial waste stream (intended for the packaging and food industries). Industrial practice shows that the analyzed group of waste belongs to the group of waste that is difficult to process. This is due to a significant degree of contamination (through metallization and printing). These layers cannot be removed by mechanical recycling, so they must be processed together with the polymer material. The conducted research allowed us to determine the impact of this type of contamination on the properties of the re-granulate in relation to the original material. The innovation of this research was the use of highly contaminated polymer materials from the industrial waste stream to obtain re-granulates that can be successfully used in new applications. The publication presents data and test results for two types of re-granulates obtained from polypropylene films. The first group of granulates was the primary polymer that was used for the production of cast films. The second group was re-granulated waste made from densely overprinted and metallized polypropylene film.

## 2. Materials and Methods

The subjects of this research are two types of polymer materials: primary PP granulate and post-industrial re-granulate (PIR). The primary material was used to produce films using the casting method, which was metallized in the next stage. The waste came from the manufacturer of metallized polypropylene packaging film. Through Granulat-Bis Comp., waste (metallized foils) was obtained directly from their producer. In the next technological step, the obtained post-industrial waste was further processed. The aim of this work was to fragment the foil and then agglomerate it. The last stage was re-granulation in order to obtain repeatable re-granulation (in terms of its shape and properties). However, due to the confidentiality clause and the manufacturer’s know-how, it was not possible to provide the source of the raw material or the exact type of polymers used. The original material (PP_cast) and re-granulate (rPP_metallized) were obtained from Granulat-Bis (Czestochowa, Poland) in order to produce standardized samples and analyze and compare their properties.

### 2.1. Tested Materials

The properties of the primary granulate (used to produce the film using the casting method), as well as the re-granulate and agglomerate (derived from the recycled overprinted metallized film), were analyzed as part of the work (Figure 2 and Table 1).

The films from which the re-granulate was made were waste materials from the casting with overprinting and metallizing process. Films of this type are multi-layer materials featuring the structure shown in Figure 2.

The photos were taken using a VHX 700 digital optical microscope Keyence Ltd., (Osaka, Japan). Microscopic observations revealed the existence of the following three layers: an intermediate layer, a barrier layer, and a metallized layer (the ABA-type film). After the production cycle, the process residues and waste film were subjected to a thickening (agglomeration) process and then, re-granulation.

### 2.2. The Process of Preparing Test Samples

Post-industrial waste films were produced by pouring a liquid polymer through a slotted head onto a calendar roll (cast films). The metallized films that were recycled were post-production waste. The polymer re-granulate was produced from the obtained cast films in the next few steps. This re-granulate was used to produce standardized moldings for further testing.

The obtained metallized food wrap film was comminuted and thickened using a Plastcompactor HV device from Herbold Meckesheim GmbH (Meckesheim, Germany).

This step was necessary in order to correctly and repeatably dose the recycled material into a RrecoSTAR dynamic 85C-VAC granulating extruder from Starlinger & Co GmbH (Vienna, Austria). The obtained agglomerates were then processed into re-granulate using a single-screw cascade extruder. As a result of the extrusion and granulation process, the polymer re-granulate was obtained (Figure 3 and Figure 4).

The re-granulate was produced, and the primary material was then processed using a Battenfeld SmartPower 60 hydraulic injection molding machine Wittmann Battenfeld GmbH (Kottingbrunn, Austria) to produce type “A1” test samples intended for the further testing of mechanical and structural properties.

The samples were made in accordance with the standard ISO 294-1 [29]. The samples were prepared using the following processing conditions: a melt temperature of 230 °C, injection pressure of 450 bar, injection speed of 75 mm/s, holding pressure of 120 bar, and mold temperature of 4 0 °C.

### 2.3. Measuring Equipment Used 

An analysis of differences in density of the materials subject to comparison was used to compare basic physical properties. Determining sample density is an important element of the quality control of raw materials and finished products. 

The density measurement was carried out for fragments taken from standardized samples produced by injection. The density measurement was carried out in accordance with the ISO 1183-1 standard [30]. A measuring station with the AS 220.3Y laboratory scale Radwag company (Radom, Poland) was used for the density measurements. 

The determination of the bulk density of the granulates and agglomerates was made in accordance with the PN-EN ISO 60 standard [31]. For this purpose, a special measuring station was used, which was made in accordance with the standard. The bulk density of each material (granulate, agglomerate, and re-granulate) was measured.

The determination of the processing factor expressed as the melt flow index was made in accordance with the PN-EN ISO 1133 standard [32]. One of the basic technological properties is the examination of the mass and volume flow rates of both samples. An LMI 5000 capillary plastometer was used in the research of the Dynisco Europe GmbH company (Heilbronn, Germany). The measurements used a weight of 2.16 kg and a measurement temperature of 230 °C. This analysis was performed to verify the impact of impurities in the form of unfiltered overprint residues and metallization on the technological indicators, viscosity, and flowability of the re-granulates produced. The measured samples of re-granulates weighing up to 10 g were dried to remove any residual moisture using a Max50 laboratory moisture analyzer, Radwag. Company (Radom, Poland). The mass flow rate was determined for 5 measurement sections in a single measurement. The average value for 5 measurements was given as the measurement result.

To determine the content of impurities, a method based on the ISO 3451-1:2019 ‘Plastics, Determination of ash, Part 1: General methods’ was used [33]. The initial tests included ashing the materials in an SNOL company (Utena, Lithuania) muffle furnace at a temperature of 600 °C for 30 min in order to assess the content of solid residues. The temperature was limited to 600 °C to prevent the melting of the aluminum layer of the film. 

The analysis of the mechanical properties of the standardized samples made from the re-granulates obtained from the waste PP films was carried out in accordance with the ISO 527-1 standard [34]. The universal testing machine type AGX-V with a 50 kN force cell by Shimadzu Comp. (Kyoto, Japan) was used in the tests. The samples were stretched at a speed of 50 mm/min. Three repetitions were performed for each type of material.

The Charpy impact strength analysis of the standardized test samples was obtained from the re-granulates—the measurement was made on a notched sample. In order to determine changes in impact strength, an analysis was carried out in accordance with the ISO 179-1 standard [35]. The tests used standard samples with a type “A1” notch cut, 2 mm deep, with an angle of 45° and a rounding radius of 0.25 mm. A Zwick/Roell (Ulm, Germany) notch cutter was used to make the notch. The impact strength test was carried out using a Zwick/Roell Hit 5.5P (Ulm, Germany).

In order to verify the mechanical properties, tests were also carried out to determine the impact of impurities from overprinting and metallization on changes in the hardness of the standardized samples made from the obtained re-granulates. The hardness tests were performed in accordance with the PN-EN ISO 2039-2 standard [36]. A Sinowon DigiRock DP3 ball hardness tester was used for this purpose (Dongguan, China). A steel ball with a diameter of 6.350 mm and a test force of 980.7 N was used. The load application time was 15 s.

The analysis of the basic thermal properties, including the melting point, crystallization temperature, and melting enthalpy, was carried out. The tests were carried out in accordance with the PN-EN ISO 11357 standard [37], using a POLYMA 214 differential scanning calorimeter from Netzsch (Selb, Germany). Samples taken from the finished products (standardized moldings obtained in the injection process) were placed in aluminum measuring crucibles. The heat–cool–heat (H-C-H) temperature program was used for the measurement. An inert atmosphere was used in the measurement cell—it was flushed with nitrogen. The temperature range used was from −20 °C to + 200 °C.

The analysis of dynamic mechanical properties was carried out in accordance with the PN-EN ISO 6721-1 standard [38]. A Netzsch DMA device was used in the research (Selb, Germany). The dynamic mechanical properties of both types of materials were analyzed during the three-point bending test. Bars sized (4 mm × 10 mm, × 55 mm) were used as samples. The tests were carried out in the measurement range from −20 °C to + 80 °C.

Microscopic analysis was carried out for the recycled films obtained from re-granulates and standardized parts. The microscopic analysis of the samples was performed using a VHX 700 electronic optical microscope Keyence Ltd. (Osaka, Japan). 

## 3. Results

### 3.1. Properties of Granulate, Agglomerate and Re-Granulate

Properties such as density, bulk density, mass flow rate and impurity content PP cast granules, agglomerate and re-granulate were analyzed, Table 2.

Granulate density tests were carried out for both types of samples (PP_cast and rPP_metallized granules) and the agglomerate as an intermediate product [30]. As density measurements showed, the difference between the material marked as a PP cast and the rPP_metallized material was only 0.78%, which is a negligible value. However, a significant difference was recorded for the bulk density value. For the material marked as PP_cast, the bulk density value was 6.28% higher than that of re-granulate (rPP_metallized). The recorded difference was probably due to the reprocessing and re-granulation process [39].

As shown by the analysis of the mass flow rate, significant differences in the value of this parameter were recorded. Re-granulation containing impurities in the form of metallization residues is characterized by a mass flow rate value that is approximately 36% lower than the primary material without impurities. The tests show that even a small admixture of impurities in the material can lead to a significant reduction in the processability coefficients and viscosity of the polymer. The difference in the mass flow rate for the tested re-granulates is 5.28 [g/10 min], which may significantly affect the behavior of the material during processing in future applications. 

The analysis of the solids content (ash/metal) shows that the average ash content for the primary material is approximately 0.06% of the mass of the tested samples. The weight of residues of the film made of virgin PP was over 0.8% larger. The film made of virgin PP had over 0.8% more weight of residues after ashing, which was due to the presence of aluminum. No significant reduction in solid residues was observed in the agglomeration process compared to the waste film. The form of dispersion of the residues changed. On the other hand, a solids content of 0.6% and a finely fragmented ash structure were observed for the sample filtered in the re-granulation process, which demonstrated good homogenization and fragmentation on the sieves of the process line. This can certainly have an impact on the properties of the obtained test samples. The test results are presented in Table 3.

### 3.2. Density Measurement Results

The density of the test samples produced by injection molding is presented below. Five measurement repetitions were made for each material, and the values presented in the charts are average values (Table 4).

There are slight differences in the densities given in Table 2 and Table 4 and in the tests performed for the standardized samples resulting from differences in the production processes of the test materials. The higher densities of the PP_cast injection samples result from a higher degree of compaction and a higher value of the melt flow index. The content of residuals of aluminum from metallization could increase the density as well [39]. 

### 3.3. The Results of the Analysis of the Mechanical Properties of the Tested Samples

Tensile strength tests were carried out in three replications for each of the tested samples, and then the average was drawn and presented in collective bar graphs. The results were divided into the following three groups relating to the obtained values: modulus of elasticity, the maximum deformation, and the maximum stress and strain of the tested samples. The comparison of the obtained results is presented in Figure 5 and Table 5. 

As the tests have shown, the standardized samples made from the re-granulate obtained from the metallized film have higher tensile strength and a higher longitudinal modulus of elasticity. Samples made from the PP_cast virgin material have a tensile strength that is 3.5% lower than samples made from the metal-containing re-granulate. However, the longitudinal modulus of elasticity of the samples obtained from the pure re-granulate is lower by approximately 10%.

It should be noted, however, that impurities present in the material, in the form of metal residues, have an adverse impact on the strain-to-failure ratio of the tested samples. The strain value for samples made from the rPP re-granulate is 71.14% lower than in the case of the primary samples containing no impurities [10,21].

As shown by the analysis of the mechanical properties based on the static tensile test, impurities in the form of metallization of the PP film have a positive effect on the tensile strength and the longitudinal modulus of elasticity. However, these impurities cause a significant decrease in deformability, expressed as strain at failure, which is characteristic of materials containing non-polymeric impurities.

### 3.4. Impact Strength Results

An analysis of impact strength was also carried out according to the Charpy method for the standardized samples obtained from both types of re-granulates, Table 6.

The analysis revealed slight differences in the impact strength of both tested sample types. As shown by the impact strength tests, the impact strength of samples obtained from the re-granulate containing impurities is 5.64% lower than the impact strength of samples without impurities (PP_cast).

This is confirmed by the fact that impurities from other materials not retained on the sieves in recycling processes cause a slight deterioration in impact strength, which is also a consequence of smaller tensile elongation. This is advantageous in the context of using re-granulates from industrial waste.

### 3.5. Hardness Analysis Results 

The next test was the assessment of the surface hardness using the Rockwell M method, Table 7.

The hardness analysis using the HRM method (with a ball indenter) showed slight differences in the hardness value in favor of the samples made from the rPP re-granulate. Most likely, the reason for this is that metallic inclusions remained after the recycling process in the body of the material, which resulted in an increase in the hardness value compared to the primary material from which the film was made.

### 3.6. Thermal Analysis DSC and DMA

The analysis of the thermal properties included the following three measurements: for the PP_cast granulate, for the re-granulate obtained from the metallized film (rPP_metallized), and for the agglomerate of this film as an intermediate material.

The DSC analysis of the thermal properties of the agglomerate obtained from the film was aimed at verifying thermal changes occurring in the polymer material during the intermediate technological process. It allowed the impact of thickening and agglomeration to be assessed on the thermal properties of polypropylene containing impurities in the form of a metallized layer.

For better insight, the differential scanning calorimetry curves (thermograms) were separated (parts of the graph containing isothermal areas were removed) and superimposed on each other to compare the results and their graphical presentation. Further in the article, we also present a table containing a summary of the values of characteristic temperatures obtained during the analysis by differential scanning calorimetry, Figure 6 and Figure 7 as well as Table 8 and Table 9.

It should be noted that the melting and crystallization temperatures vary depending on the type of material and its condition (agglomerate/re-granulate). Attention should also be paid to the change in melting enthalpy, which, depending on the material type, is 99.41 J/g for the primary polymer (PP_cast), 87.8 J/g for the agglomerate and 82.88 J/g for the re-granulate (rPP_metallized). The results indicate a decrease in melting and crystallization enthalpies during subsequent processing stages, which might be caused by the progressive thermomechanical degradation processes as well as the minor impact of impurities. The factor influencing the reduction in melting enthalpy in the agglomerate and re-granulate is another technological process that thermally loads the material. Moreover, attention should be paid to the method of obtaining the rPP to re-granulate itself. It is a two-stage process, where, in the first stage, there is agglomeration with the release of heat during friction between two plates, and then in the second stage, the re-granulation process uses a granulating extruder, where the heat necessary for melting and homogenization comes from a system of heaters and from friction. As confirmation of these observations, attention should be paid to the values of melting and crystallization temperatures. For the obtained values, lower values of crystallization and melting temperatures were recorded for the agglomerate and re-granulate (rPP_metallized). The decrease in these values indicates progressive degradation. Similarly, the obtained onset temperature values indicate a deterioration of the thermal properties of recycled materials. The PP_cast material is characterized by the highest onset temperature values, which proves its greater thermal stability.

Another thermal test was the dynamic, mechanical thermal analysis (DMTA) using a DMA 303 Excelsior analyzer. The test samples were taken from the injection moldings. Changes in the elastic storage modulus E’ and tan δ in the temperature range from −100 °C to 150 °C were analyzed. The tests were performed for two frequencies, 1 Hz and 10 Hz, but the thermogram shows the results only for 10 Hz for easier presentation of the results in Figure 8.

Significant differences are visible in the changes in the storage modulus E’ as a function of temperature changes, which persist throughout the entire temperature range. The samples made from the rPP recyclate-containing metal had E’ values 1.000 MPa higher at −100 °C than the samples made from the virgin PP_cast material. The reason for this may be the content of impurities, residuals of metallization, and remnants of materials that were fragmented during recycling processes and which constituted a phase in the mass of the molded parts that caused slight strengthening. The glass transition temperature values read from the E’ and tan δ curves are similar for both tested materials. In the case of the E’ curves, the difference is 2K, while in the case of the extreme tan δ, a higher value was achieved by the sample made of the primary PP_cast material. The extreme values of the mechanical loss coefficients tan δ for both samples are similar, but they differ significantly after exceeding the glass transition temperature range. The rPP sample loses its damping properties, which corresponds to a much lower tan δ value, while the PP_cast sample, after exceeding 40 °C, shows a significant increase in this parameter, which then remains at the level of approximately 0.1 in the rest of the test temperature range of 100–150 °C.

This research confirms an increase in the stiffness of the metallized recyclate with a simultaneous decrease in the damping properties (which is due to the viscoelastic properties of the virgin material). This behavior was also confirmed by the previously performed tests.

## 4. Conclusions

The research has shown that a properly carried out recycling process leads to obtaining a full-fledged product (re-granulate) that does not differ significantly from re-granulate not containing inclusions. As shown by the analysis of the mechanical properties, the obtained re-granulates and the standardized samples containing impurities (metallization residues) are characterized by better static mechanical properties, higher hardness, and slightly lower impact strength.

The tests have also shown that the content of Impurities may affect the processability of the tested polymer materials. Contamination in the form of metallization and overprinting residues leads to the deterioration of processability factors (defined by the melt flow index).

As the research has shown, the basic physical properties, such as density, do not differ significantly for the PP_cast and rPP_metallized materials. However, it has been shown that re-processing (agglomeration and re-granulation) causes a change in the bulk density of the polymer, which may affect the dosing and plasticization characteristics.

The mechanical properties determined during the tangential tensile test do not differ significantly (the difference was 1.15 MPa, which is 3.62%). However, significant differences were recorded for the strain values. The impurities contained in the rPP_metallized samples significantly reduced the strain obtained during the test (rPP_metallized samples are characterized by a 71% lower strain value).

It is extremely interesting that slight differences in the impact strength values were recorded, but it should be noted that the hardness of the rPP_metallized samples was significantly higher. The obtained result was probably influenced by impurities (metallization residue), causing an increase in this property.

Thermal tests based on the DSC method showed small differences in the melting point, crystallization temperature, and onset temperature of the PP_cast and rPP_metallized samples. However, the obtained results indicate that mechanical recycling leads to the deterioration of thermal properties; the reason for this behavior of the material is thermo-mechanical degradation. However, DMTA tests confirm a significant increase in the stiffness of samples obtained from the rPP_metallized material, which is caused by the content of impurities (metallization).

An in-depth analysis of the physical properties of the primary materials and re-granulates obtained from waste from the metallized film production process allowed us to conclude that the obtained properties do not differ significantly from the properties of the primary material to such an extent that its reuse is impossible. There is a visible impact of subsequent technological processes related to recycling (agglomeration and re-granulation) in the form of a slight deterioration of certain properties of re-granulates.

Further analysis of the behavior of recycled materials is necessary for a better and more informed selection of such feedstock for future industrial applications. The research described above confirms the full usefulness of this type of materials, and the results obtained do not reveal any limitations to their functionality in such applications.

## Figures and Tables

**Figure 1 materials-17-01739-f001:**
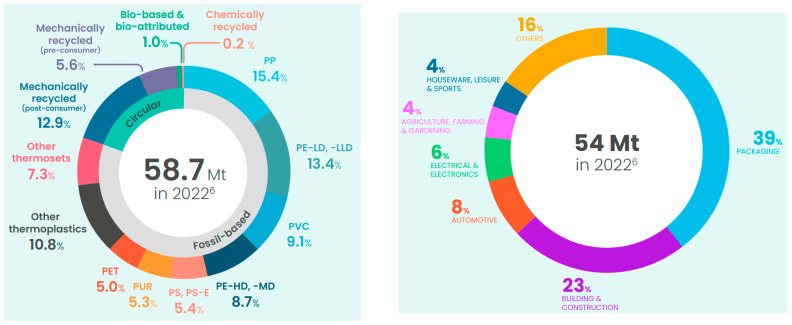
European market polymer production and conversion [18].

**Figure 2 materials-17-01739-f002:**
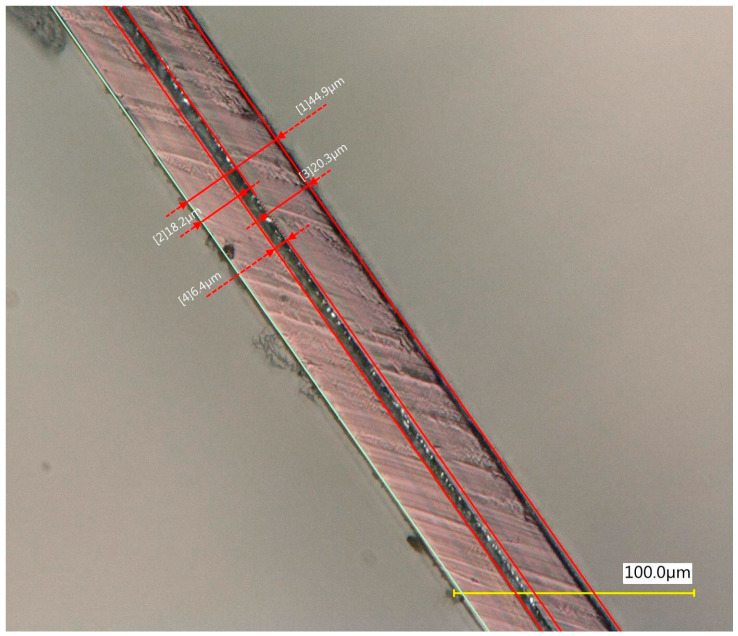
Cross-sectional structure of the tested waste film.

**Figure 3 materials-17-01739-f003:**
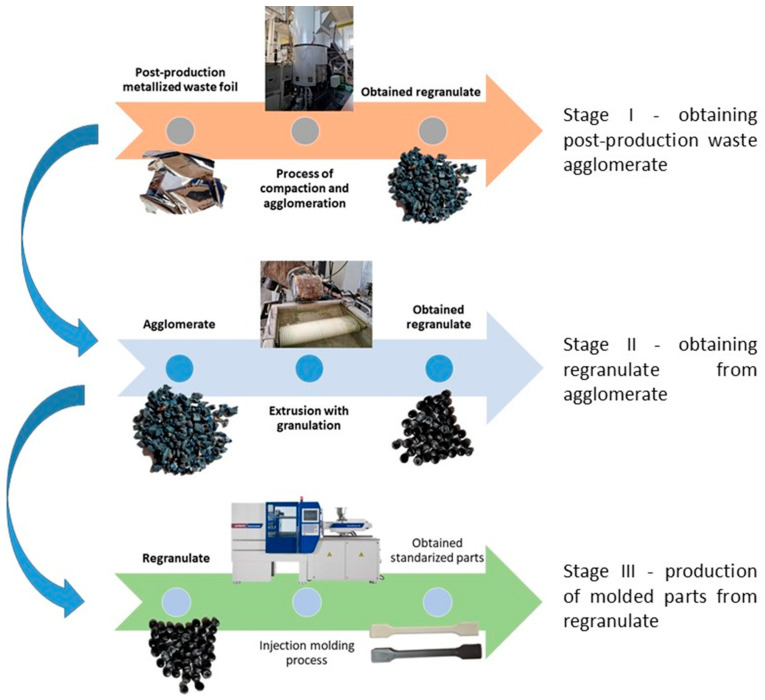
The scheme of producing agglomerate, re-granulate and standardized test samples.

**Figure 4 materials-17-01739-f004:**
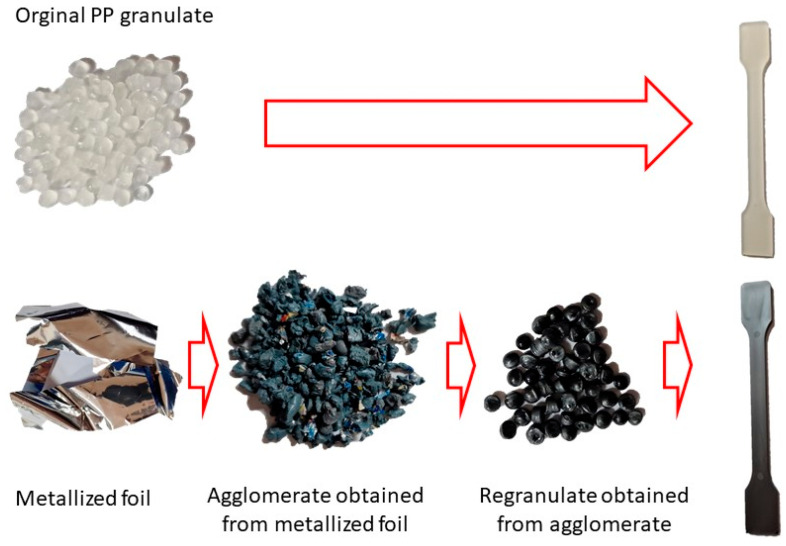
The process of standardized sample preparation.

**Figure 5 materials-17-01739-f005:**
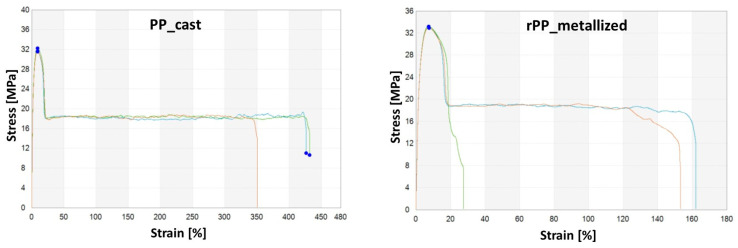
Mechanical properties analysis results.

**Figure 6 materials-17-01739-f006:**
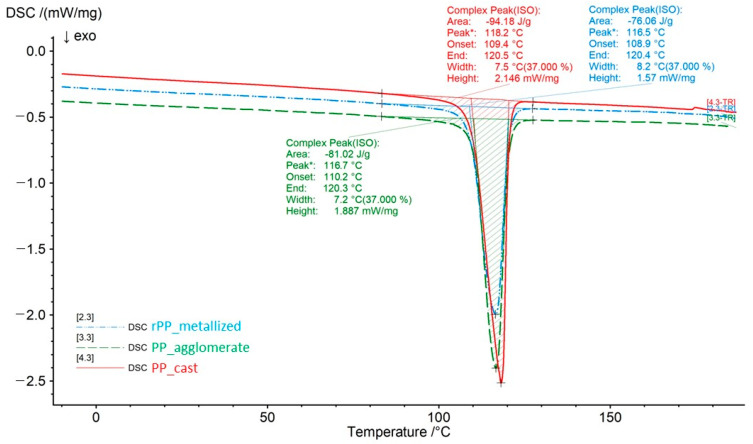
DSC thermograms of the tested samples during cooling.

**Figure 7 materials-17-01739-f007:**
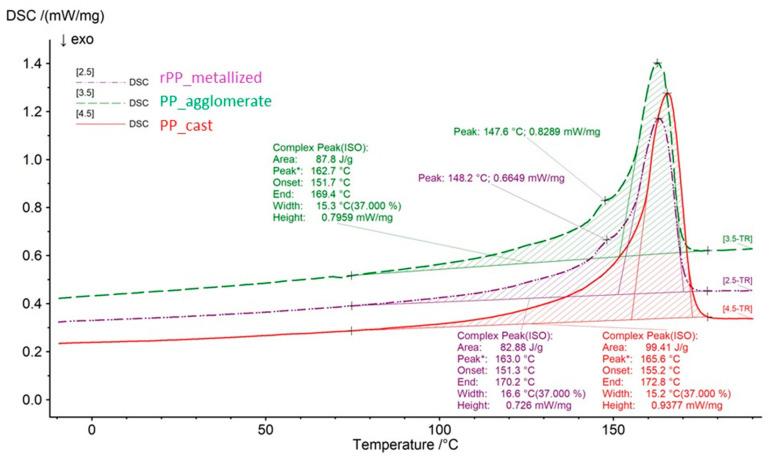
DSC thermograms of the tested samples during second heating.

**Figure 8 materials-17-01739-f008:**
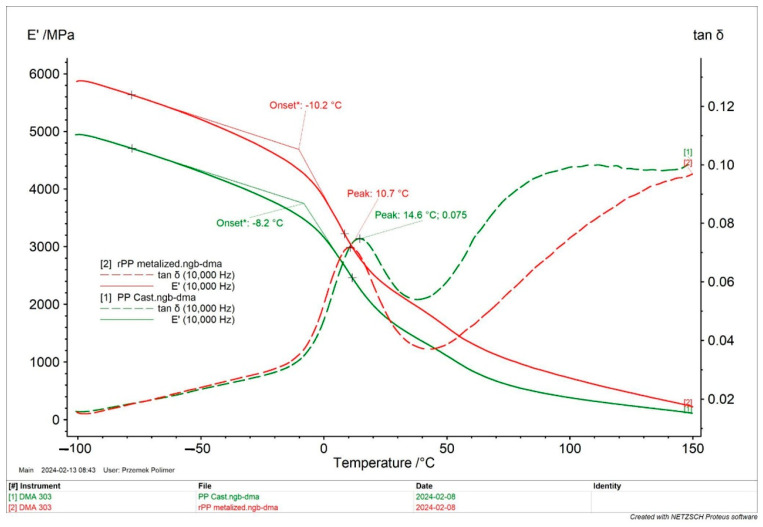
DMA thermograms (E’ storage modulus and tan δ tangent delta) at a frequency of 10 Hz, for both tested samples.

**Table 1 materials-17-01739-t001:** Description of all materials tested during the research.

Tested Materials		
Sample Name	Properties Description	View of the Sample
**1** **PP_cast** **reference sample**	The primary PP granulate is used to produce food wrap films using the casting method. Due to business confidentiality, the trade name of the material may not be disclosed.	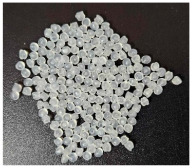
**2_1** **Metallized film**	Processed waste from metallized and overprinted film constituting the input research material	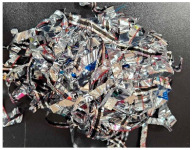
**2_2** **Agglomerate**	Agglomerate of waste metallized film as a semi-finished product for the production of re-granulate	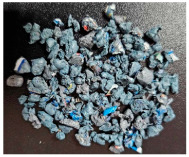
**2_3** **rPP_metallized**	PP re-granulate (rPP) obtained from metallized and overprinted film	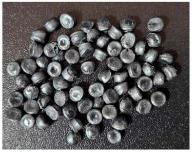

**Table 2 materials-17-01739-t002:** Basic properties of the compared polymer materials.

Parameter	PP_Cast	rPP_Metallized
Density	0.901 g/cm^3^	0.908 g/cm^3^
Density	agglomerate—0.771 g/cm^3^	
Bulk density	0.525 g/cm^3^	0.492 g/cm^3^
Bulk density	agglomerate—0.310 g/cm^3^	
Mass flow rate (MFR) (230 °C/2.16 kg)	14.55 g/10 min.	9.27 g/10 min.
Melting temperature	169.7 °C	168.6 °C
Crystallization temperature	118.2 °C	115.1 °C
Moisture content	less than 0.1% weight	

**Table 3 materials-17-01739-t003:** Ash content analysis results for granulate, agglomerate and re-granulate.

PP Virgin	Metallized/Printed Film	Agglomerate	rPP
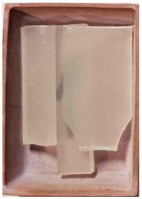	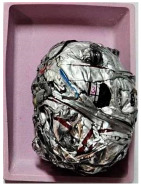	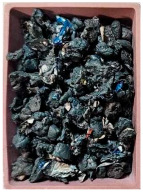	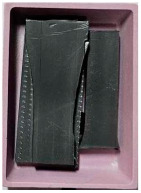
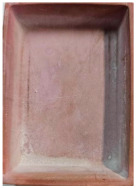	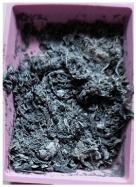	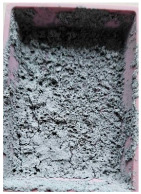	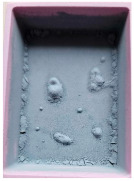
**Ash content** **0.0623%**	**Ash content** **0.8786%**	**Ash content** **0.8672%**	**Ash content** **0.6002%**

**Table 4 materials-17-01739-t004:** Statistical data regarding density measurement of the samples.

Density, g/cm^3^		
	PP_Cast	rPP_Metallized
Average value	0.904	0.907
Standard deviation	0.01	0.001

**Table 5 materials-17-01739-t005:** Summary of mechanical properties for compared samples.

	PP_Cast	rPP_Metallized
**Tensile strength, MPa**	31.78	32.93
Standard deviation	0.39	0.14
**Strain, %**	402	116
Standard deviation	45.63	79.37
**Modulus of elasticity, MPa**	1305.77	1462.15
Standard deviation	35.6	12.49

**Table 6 materials-17-01739-t006:** Statistical data of impact strength of the comparison samples.

Impact Strength, kJ/m^2^		
	PP_Cast	rPP_Metallized
Average value	3.08	2.91
Standard deviation	0.08	0.09

**Table 7 materials-17-01739-t007:** Statistical data of Rockwell hardness [HRM].

Rockwell Hardness, HRM		
	PP_Cast	rPP_Metallized
Average value	79.44	89.8
Standard deviation	0.61	2.37

**Table 8 materials-17-01739-t008:** Thermal properties of the analyzed materials—crystallization.

	Crystallization Temp., °C	Crystallization Enthalpy, J/g	Onset Temp., °C
PP_cast	118.2	99.18	109.4
Agglomerate from PP film	116.7	81.02	110.2
rPP_metallized	116.5	76.06	108.9

**Table 9 materials-17-01739-t009:** Summary of thermal properties of the analyzed materials—second heating.

	Melting Temp., °C	Melting Enthalpy, J/g *	Onset Temp., °C
PP_cast	165.6	99.41	155.2
Agglomerate from PP film	(147.6) 162.7	87.80	151.7
rPP_metallized	(148.2) 163.0	82.88	151.3

* The value obtained comes from the second heating cycle and was read from the heating curve.

## Data Availability

Data are contained within the article.

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
