# Peer review of "Analysis of Mechanical and Thermal Properties of Polymer Materials Derived from Recycled Overprinted Metallized PP Films"

_materials, 2024, doi:10.3390/ma17081739_

Round 1
Reviewer 1 Report
Comments and Suggestions for Authors
This manuscript deals with very interesting theme, that is recycling of the polypropylene film material. However, there are some minor issues that need to be address before publication. The Introduction part is quite long. At the end of this part the highlight it missing; authors must emphasize the main goal of this investigation. On the Fig. 3 on the page 8, there is mistake in spelling “original”; please correct it. On the page 9, line 263, regarding the analysis of the mechanical properties authors mentioned 5 repetitions performed for each type of material. However, on the page 10 in the 3.1. The results of the analysis of the mechanical properties of the tested samples authors are suggesting that tests were carried out in 3 replications. What is the correct number? The Figs 4, 7 and 8, as well as Tables 4, 6 and 7 are totally needlessly. Authors are advised to summarize all data from these Figs and Table into only one Table. Regarding the DSC results, the authors focused only on the enthalpy (lines 421-425). What about the melting and crystallization temperatures? In the Table 3 (by the way, check the correct numbering of the tables!), values of the corresponding temperatures are showed as peak temperatures. However, it is well known fact that Tonset temperature is the more precise, since it is independent on sample mass. If we check the onset values on the Fig. 10 we can notice the difference in the values of the different samples. This fact need the proper comment! Concerning the DMA analysis, authors are constantly labelling the tan δ as the “tgd” or “TgD”. Please correct it! Likewise, where is the curve of the loss modulus (E”)? Finally, authors are advised to perform other thermal analysis, like thermogravimetric analysis (TGA) in order to gain more information about possible effect of the recycling treatment on the thermal stability of these materials.
Comments on the Quality of English LanguageMinor editing of English language required.
Author Response
Dear Reviewer
At the outset, I would like to thank you for the prepared review. Thank you very much for the extensive study containing such a large amount of valuable information, which will significantly influence the quality not only of the submitted publication but also of future work. I agree with all the comments .
Best regrads
Tomasz Stachowiak

Reviewer 2 Report
Comments and Suggestions for Authors
This study analyzed properties of the primary material used to produce film using the casting method in comparison with the industrial recyclate obtained by the processing of film made of the primary material and then overprinted and metallized. The process of obtaining re-granulates and preparing test samples was presented, and the mechanical, structural, and thermal properties of the tested materials were compared. I recommend publication of this study, however, also suggest the following minor revisions.
1. The stress-strain curves by the mechanical testing should be depicted in addition to data in Table 5 to understand profiles of mechanical properties more clearly.
2. The conclusion is too abstractive, which should be revised as it expresses significant findings of this study more specifically.
Comments on the Quality of English LanguageModerate editing of English language is required.
Author Response
At the outset, I would like to thank you for the prepared review. Thank you very much for the extensive study containing such a large amount of valuable information, which will significantly influence the quality not only of the submitted publication but also of future work. I agree with all the comments.
Best regards
Tomasz Stachowiak

Reviewer 3 Report
Comments and Suggestions for Authors
Dear Authors
Your article covers an attractive topic, characterization of selected properties on recycled metallized polymers.
However, the article is not written in a scientific way. It needs an improvement. Namely, the title should be more specific, of course if you agree.
The abstract needs also addition of the most/best achieved results.
the Introduction part needs a big improvement with proper citation ...This is you state of the art and needs to give a brief idea....what is the status of the science of this topic...Comparison of the current achievements, etc...
The Experimental part is not well written...some results are mixed or interpret in this section that is not allowed.
Result section needs improving as well. The interpretation of the obtained data could be organized in a better way. Sometimes there are repetitive presentation of data, e.g. in Table and Figure for the same data set?
Finally, the Conclusion should briefly summarize the most important results and clearly says the main achievements. Do the objectives/aims of the study are achieved?
Thank you for understanding my comments/suggestions. In attachment you can find most of these suggestions written in your manuscript. Thank you.

It could be improved.
Author Response
At the outset, I would like to thank you for the prepared review. Thank you very much for the extensive study containing such a large amount of valuable information, which will significantly influence the quality not only of the submitted publication but also of future work. I agree with all the comments .
Best regards
Tomasz Stachowiak

Reviewer 4 Report
Comments and Suggestions for Authors
This paper studies the properties of the main materials used to produce thin films by casting method, and compares them with the industrial recycled materials obtained by processing and then overprinting and metallization of the films made of the main materials. It mainly compares the mechanical properties, structural properties and thermal properties of the tested materials, which has certain research value, but some modifications are needed before publication:
1. The abstract of this paper needs to be re-written. It is hoped that the previous background introduction should be brief, and it is important to introduce the research content, research methods, research results (especially relevant research data) and research significance of this paper. Please revise it carefully.
2. The first reference in the first paragraph of the introduction is [7-12], which is obviously inappropriate. Shouldn't we start with [1]? Moreover, it is hoped that the author should not use the collective quotation at one time, and it is suggested to use it separately to enhance the persuasiveness of this paper, and it is hoped that the effect of careful revision;
3. Please add the last paragraph in the Introduction to summarize the differences between this paper and previous literature studies, so as to reflect the value of this study. Please revise it.
4. Tables 4, 6, 7 and Figures 4, 7 and 8 in the text need to be redesigned to increase readability and aesthetics, and the font size should be consistent with other figures;
5. The format of references in this paper is inconsistent, especially the DOI and URL links, please unify them
6. It is suggested that the conclusion should be described by points instead of sections. In addition, the conclusion should be simplified and focused.
Author Response

(The authors gave the same response as above.)

Reviewer 5 Report
Comments and Suggestions for Authors
I find that the paper entitled: "Analysis of selected properties of polymer materials derived from recycled overprinted metallized films" is a very interesting. According to statistical data from Plastics Europe, approximately 40 % of processed thermoplastics are used to produce packaging, including single- and multi-layer film packaging. Growing requirements and new EU directives require the use of recycled materials in new products, which is not easy because the properties of recyclates may differ significantly from those of the primary materials with which the former are mixed. Polymer plastics and their composites are one of the most frequently used materials in the packaging and food industries: disposable and reusable packaging, layered films with barrier properties, as well as densely overprinted polymer films and metallized food wrap films. Their disposal is a major environmental problem.
In this paper was presented the research of properties of the primary material used to produce film using the casting method in comparison with the industrial recyclate obtained by the processing of film made of the primary material and then overprinted and metallized. The process of obtaining re-granulates and preparing test samples was presented, and the mechanical, structural and thermal properties of the tested materials were compared.
The research has shown that a properly carried out recycling process leads to obtaining a full-fledged product (regranulate) that does not differ significantly from regranulate containing no inclusions. As shown by the analysis of the mechanical properties, the obtained regranulates and the standardized samples containing impurities (metallization residues) are characterized by better static mechanical properties, higher hardness and slightly lower impact strength. The tests have also shown that the content of impurities may affect the processability of the tested polymer materials. Contamination in the form of metallization and overprinting residues leads to deterioration of processability factors (defined by the melt flow index). An in-depth analysis of the physical properties of the primary materials and regranulates obtained from waste from the metallized film production process allows us to conclude that the obtained properties do not differ significantly from the properties of the primary material to such an extent that its reuse would be impossible. There is a visible impact of subsequent technological processes related to recycling (agglomeration and regranulation) in the form of a slight deterioration of certain properties of regranulates.
I support future work involving analysis of the behavior of recycled materials which necessary for a better and more informed selection of such feedstock for future industrial applications.
I agree that research described confirms full usefulness of this type of materials and the results obtained do not have not revealed any limitations on their functionality in such applications.
Accordingly, I recommend the accept of paper in present form.
Author Response
Dear Reviewer
At the outset, I would like to thank you for the prepared review. Thank you very much for the extensive study containing such a large amount of valuable information, which will significantly influence the quality not only of the submitted publication but also of future work. I agree with all the comments.
Best regards
Tomasz Stachowiak

Round 2
Reviewer 3 Report
Comments and Suggestions for Authors
Dear Authors,
Your manuscript is an improved version now. However it is a room for further improvement in order to be better version for readers.
Please find attached version of the manuscript with my suggestions in it. Thank you.

English could be improved.
Author Response
Dear Reviewr
At the outset, I would like to thank you for the prepared review. Thank you very much for the extensive study containing such a large amount of valuable information, which will significantly influence the quality not only of the submitted publication but also of future work. I agree with all the comments.
Based on comments the title of the publication was changed (second change)
Analysis of mechanical and thermal properties of polymer materials derived from recycled overprinted metallized PP films
Best regards
Tomasz Stachowiak
